# Oxidative Stress and Inflammation Biomarkers in Postoperative Pain Modulation in Surgically Treated Patients with Laryngeal Cancer—Pilot Study

**DOI:** 10.3390/cells12101391

**Published:** 2023-05-14

**Authors:** Katarina Savic Vujovic, Andjela Zivkovic, Ivan Dozic, Andja Cirkovic, Branislava Medic, Dragana Srebro, Sonja Vuckovic, Jovica Milovanovic, Ana Jotic

**Affiliations:** 1Department of Pharmacology, Clinical Pharmacology and Toxicology, Faculty of Medicine, University of Belgrade, Dr Subotica 1, 11129 Belgrade, Serbia; katarinasavicvujovic@gmail.com (K.S.V.); brankicamedic@gmail.com (B.M.); srebrodragana1@gmail.com (D.S.); vuckovicsonja1@gmail.com (S.V.); 2Faculty of Medicine, University of Belgrade, Dr Subotica 1, P.O. Box 38, 11129 Belgrade, Serbia; zivkovi.angela340@gmail.com (A.Z.); jmtmilov@gmail.com (J.M.); 3Department of Pathology, School of Dental Medicine, University of Belgrade, Dr Subotica-Starijeg 1, 11000 Belgrade, Serbia; ivan.dozic@stomf.bg.ac.rs; 4Institute for Medical Statistics and Informatics, Faculty of Medicine, University of Belgrade, 11129 Belgrade, Serbia; andja.cirkovic@med.bg.ac.rs; 5Clinic for Otorhinolaryngology and Maxillofacial Surgery, University Clinical Center of Serbia, Pasterova 2, 11000 Belgrade, Serbia

**Keywords:** postoperative pain, tramadol, NSAID, oxidative stress, SOD, MDA

## Abstract

(1) Background: Surgical treatment of laryngeal carcinoma includes different types of laryngectomies with neck dissection. Surgical tissue damage triggers an inflammatory response, leading to the release of pro-inflammatory molecules. This increases reactive oxygen species production and decreases antioxidant defense mechanisms, leading to postoperative oxidative stress. The aim of this study was to assess the correlation between oxidative stress (malondialdehyde, MDA; glutathione peroxidase, GPX; superoxide dismutase, SOD) and inflammation (interleukin 1, IL-1; interleukin-6, IL-6; C–reactive protein, CRP) parameters and postoperative pain management in patients surgically treated with laryngeal cancer. (2) Methods: This prospective study included 28 patients with surgically treated laryngeal cancer. Blood samples were taken for the analysis of oxidative stress and inflammation parameters before the operative treatment and after the operative treatment (1st postoperative day and 7th postoperative day). The concentrations of MDA, SOD, GPX, IL-1, IL-6, and CRP in the serum were determined by coated enzyme-linked immunosorbent assay (ELISA). The visual analog scale (VAS) was used for pain assessment. (3) Results and conclusion: There was a correlation between oxidative stress and inflammation biomarkers and postoperative pain modulation in surgically treated patients with laryngeal cancer. Age, more extensive surgery, CRP values, and use of tramadol were predictors for oxidative stress parameters.

## 1. Introduction

Laryngeal carcinoma accounts for 1–3% of all head and neck malignancies [1]. Surgical treatment of laryngeal carcinoma includes different types of more or less extensive laryngectomies with reconstructions of the aerodigestive tract with local or remote flaps and associated unilateral or bilateral neck dissection [2]. Surgical cordectomy and different forms of partial laryngectomy are considered less extensive operations. Total laryngectomy or pharyngolaryngectomy with local or distant flap reconstructions are considered major head and neck surgery [2]. Neck surgery was also associated with increased risk of higher intensity of postoperative pain and other symptoms like dysesthesia, paresthesia, and hypersensitivity, due to the interruption of the nerve fibers. [3]. These extended procedures are considered to be the most painful surgeries in otorhinolaryngology and require optimal postoperative care. Intense postoperative pain is usually inadequately controlled and significantly affects the recovery of these patients [4,5].

Oxidative stress is a condition caused by an imbalance between the production and accumulation of reactive oxygen species (ROS) and the body’s ability to detoxify these products [6,7]. An excess of ROS causes damage to all cellular components. The process of oxidative stress causes the peroxidation of lipids and proteins, the formation of lipid peroxides, DNA fragmentation, and the development of cell death. ROS activity is usually assessed indirectly by measuring stable products derived from the interaction of these radicals with cellular components. Malondialdehyde (MDA) is a breakdown product of lipid peroxides, belongs to the group of aldehydes, and is used for the quantification of lipid peroxidation in the obtained tissue samples [6]. The body activates defense mechanisms composed of antioxidants to prevent cellular damage and suppress the destructive effects of oxidants. The most-studied cellular antioxidants are superoxide dismutase (SOD) and glutathione peroxidase 1 (GPX1). These enzymes provide the first line of defense against tissue damage caused by ROS. As part of the antioxidant pathway, SOD accelerates the conversion of superoxide to H_2_O_2_, while catalase and GPX convert H_2_O_2_ to water [6]. MDA, SOD, and GPX can be measured in serum or plasma to establish the levels of oxidative stress and antioxidant capacity of the body [6,7].

Surgical trauma can cause significant physiological stress in the body. Mechanical tissue damage results in activation of complement, polymorphonuclears, and macrophages, causing increased release of ROS and oxidative damage in the postoperative course [8,9]. Hypoxia also increases the release of endothelin-1 and the scavenging of NO, producing vasoconstriction, further hypoxia, ischemia, and subsequent release of tumor necrosis factor-α (TNFα), which serves to augment the stress response [10]. More intense oxidative damage is expected with more extensive surgical tissue damage, which can also cause a reduction in the antioxidant capabilities of the patient [11]. Oxidative stress may also alter nociception, which may cause hyperalgesia with local oxidant mechanisms. Similarly, release of ROS due to tissue damage and inflammation might increase the stimulation of sensory neurons that play a role in the transmission of pain [12,13]. Due to this type of connection, systemic markers of oxidative stress such as MDA, SOD, and GPX could serve as new biomarkers of pain intensity and postoperative pain management in surgically treated patients [8,9,10,11].

Interleukin-1 (IL-1) and interleukin-6 (IL-6) are pro-inflammatory cytokines that are released in response to tissue damage, inflammation, and infection. Their role in management of postoperative pain and oxidative stress is still being researched. During surgery, the body’s immune system triggers the release of these inflammatory cytokines. By recruiting additional immune cells and activating oxidase enzymes, IL-1 and IL-6 can additionally stimulate the production of ROS and further exacerbate oxidative stress [14]. IL-1 and IL-6, released during inflammatory responses after tissue damage, contribute to the development of hyperalgesia [15,16,17,18,19]. C-reactive protein (CRP) is an acute inflammation-phase protein and a part of the innate immune system which generates pro-inflammatory cytokines and enhances the inflammatory response [20].

The aim of this study was to assess the correlation between biomarkers of oxidative stress (MDA, SOD, and GPX) and inflammation (IL-1, IL-6, CRP) and postoperative pain management in surgically treated patients with laryngeal cancer.

## 2. Materials and Methods

### 2.1. Patient Selection

A prospective study included 28 patients with surgically treated squamocellular carcinoma of the larynx in the period from October 2022 to February 2023 at the tertiary referral center. This study was approved by the Institutional Ethics Committee (745/5-22), and all patients signed the informed consent form prior to their inclusion in the study. The study was registered at ClinicalTrials.gov (ID NCT05857202). The diagnosis of laryngeal carcinoma was confirmed by otorhinolaryngological clinical examination and laryngomicroscopic examination of the larynx with the biopsy and histopathologic examination of the tissue. Additional diagnostics (chest radiography, computed tomography of the neck, and ultrasonography of the abdomen) were performed to determine the TNM stage of the disease. The study included patients with all stages of operable laryngeal carcinoma (T1–T4, N0–N3), without previous treated malignancies. Exclusion criteria were inoperable malignant disease, presence of distant metastases, previously treated malignancies, presence of neurological or other severe comorbidities which prevent surgical treatment, the presence of neurological or other severe physical and metabolic comorbidities, substance abuse, and the inability to provide informed consent.

The modality of treatment for every patient was decided on by the institutional Oncological Board (consisting of a radiotherapist, head and neck surgeons, an oncologist, and a histopathologist). Open surgical treatment involved resection of the tumor (cordectomy or partial or total laryngectomy) with or without some form of the neck dissection in the case of cervical lymphadenopathy. Demographic, clinical, and histopathological characteristics (age and gender, tobacco use, histopathological tumor grade, TNM classification, and therapy modality) were noted.

### 2.2. Pain Assessment

The visual analog scale (VAS) was used for pain assessment [21]. Scores were based on self-reported measures of pain severity recorded with a mark placed at one point along the length of a 10 cm line that represents a continuum between the two ends of the scale (0 cm on the left end of the scale marks “no pain” and 10 cm on the right end of the scale marks “the worst pain”). Pain assessment was performed before the operative treatment and two times after the operative treatment (on the 1st postoperative day and on the 7th postoperative day). VAS scores from 0 to 4 mm were considered no pain; 5 to 44 mm, mild pain; 45 to 74 mm, moderate pain; and 75 to 100 mm, severe pain [22].

### 2.3. Postoperative Analgesia

After evaluating patient’s pain using the VAS scale, analgesic was prescribed based on the degree of pain. According to recommendations [23,24,25], mild postoperative pain was treated with non-opioid analgesics such as acetaminophen or NSAIDs, either alone or as a component of multimodal analgesia depending on clinical assessment and pain severity. For management of moderate pain, opioids (tramadol) were used alone or as a component of multimodal analgesia with non-opioid analgesics.

### 2.4. Measurement of the Oxidative Stress and Inflammatory Parameters

For the analysis of oxidative stress parameters and inflammation parameters, blood samples were taken from the patients before the operative treatment and after the operative treatment (1st postoperative day and 7th postoperative day). The concentrations of the inflammatory parameters interleukin 1 (IL-1) and 6 (IL-6) and C-reactive protein (CRP) and oxidative stress parameters glutathione peroxidase 1 (GPX1), superoxide dismutase (SOD), and malondialdehyde (MDA) in the serum were determined by coated enzyme-linked immunosorbent assay (ELISA) kits, according to the manufacturer’s instructions (Elabscience, Wuhan, China). The ELISA kits for determination of concentrations of IL-1 and IL-6 were based on the Sandwich ELISA principle, with plates pre-coated with an antibody specific to human cytokines. The optical density (OD) was measured spectrophotometrically at 450 nm, using a Multiskan EX plate reader (Thermo Fisher Scientific, Vantaa, Finland). The concentration of tested samples was calculated by comparing the OD of the samples to the standard curve created with GraphPad Prism 9.0 software (GraphPad Software Inc., San Diego, CA, USA).

### 2.5. Statistical Analysis

Categorical data was described by absolute and relative numbers, while numerical data was reported as arithmetic mean and standard deviation or median and interquartile range (IQR), depending on the data distribution. The normality was evaluated using mathematical (Shapiro–Willk, skewness and kurtosis, and coefficient of variation) and graphical (histogram, box plot) methods. For the evaluation of changing of pain intensity, parameters of oxidative stress, and biomarkers of inflammation, a Friedman test was applied with a Wilcoxon signed-rank test as a post hoc testing method. For analyzing the association between pain intensity (VAS), parameters of oxidative stress, and biomarkers of inflammation, Spearman’s rank correlation coefficient was used, because the variables did not have a normal distribution. In order to evaluate all possible factors that influence the level of oxidative stress, expressed as SOD and MDA levels, linear regression analysis (enter method) was performed, reporting the regression coefficient B, the 95% confidence level (CI) of B, and the *p* value. Univariate analysis was performed first, and all significant factors were combined in multivariate models. All statistical methods were considered significant if the *p* value was less than or equal to 0.05. The analysis was performed in IBM SPSS ver. 26.

## 3. Results

### 3.1. Patients’ Characteristics

A total of 28 patients who underwent laryngectomy were enrolled in this pilot study. The average age was 67.04 ± 6.60 years, with a majority of males (82% vs. 18%). Almost all included patients were smokers (86%). The most common histological tumor grade was G2 (20 patients, 71.4%). Early laryngeal carcinoma (T1 and T2) was detected in 53,5% of the patients, and advanced laryngeal carcinoma (T3 and T4) in 46.5% of the patients. Most of the patients were without regional nodal spread (23 patients, 82.2%). Four types of surgery were performed: cordectomy, partial laryngectomy, total laryngectomy, and laryngectomy with neck dissection. The most common was cordectomy, in 36% of all cases. Baseline and clinical characteristics of included patients are presented in Table 1.

### 3.2. Pain Intensity and Postoperative Analgesia

VAS scores that assessed pain intensity preoperatively, on the 1st postoperative and on the 7th postoperative day, are presented in Figure 1. The pain intensity significantly changed from preoperative period (med = 5, IQR 0–10) comparing to the 7th postoperative day (med = 10, IQR 0–10) (*p* < 0.001). Pain significantly increased from preoperative to the 1st postoperative day (med = 40, IQR 10–60) (*p* < 0.001) and significantly decreased from the 1st to the 7th postoperative day (*p* < 0.001).

Patients in our study complained of mild or moderate pain. There were no patients who had severe pain. NSAIDs (ketorolac or metamizole) were used for treatment of mild pain. If the patient complained of pain after NSAID administration, acetaminophen was added to the therapy. If the patient had moderate pain, tramadol was used for pain therapy. If tramadol did not relieve the pain, NSAIDs were added. NSAIDs (ketorolac or metamizole) were used alone in 13 patients (46.4%). Ketorolac or metamizole were combined with acetaminophen in 3 patients (10.7%) and with tramadol in 13 patients (28.6%). Tramadol was used alone in 4 patients (14.3%). Among non-opioid analgesics, ketorolac was the most often prescribed (19 patients, 67.9%). Prescribed analgesics are presented in Figure 2.

### 3.3. Parameters of Oxidative Stress

Median values of oxidative stress parameters preoperatively, on the 1st, and on the 7th postoperative day are presented in Figure 3. SOD and MDA parameters of oxidative stress significantly changed from preoperative to the 7th postoperative day (*p* < 0.001). SOD significantly increased between preoperative (med = 85.25, IQR 72.87–89.25) and the 1st postoperative day (med = 101.00, IQR 83.62–120.75) (*p* < 0.001) and between preoperative and the 7th postoperative day (med = 108.25, IQR 96.62–131.25) (*p* < 0.001). There was no significant change in SOD level between the 1st and the 7th postoperative day. MDA significantly increased between preoperative (med = 57.74, IQR 45.74–78.75) and the 1st postoperative day (med = 72.99, IQR 61.91–109.66) (*p* < 0.001) and between preoperative and the 7th postoperative day (med = 74.16, IQR 62.33–98.66) (*p* < 0.001). There was no significant difference between the 1st and the 7th postoperative day (*p* = 0.782). Values of GPX did not change significantly postoperatively, compared to their values preoperatively (*p* = 0.898).

### 3.4. Inflammation Parameters

Though there was an increase in values of IL-1 on the 1st and on the 7th postoperative day (med = 4.60, IQR 3.89–5.59 and med = 5.00, IQR 4.36–5.57, respectively) compared to those preoperatively (med = 4.52, IQR 3.96–5.17), the difference was not significant (*p* = 0.331). Values of IL-6 increased on the 1st postoperative day (med = 31.44, IQR 23.86–43.74), then decreased (med= 29.00, IQR 25.81–34.42) on the 7th postoperative day, but were still elevated compared to those preoperatively (med = 27.39, IQR 23.33–44.86). Changes were not statistically significant (*p* = 0.565). CRP also did not show any significant changes in the postoperative period (*p* = 0.156).

### 3.5. The Correlation between Pain Intensity and Oxidative Stress and Inflammation Parameters

Negative moderate correlation between IL-1 and GPX (ρ = −0.378, *p* = 0.047) and between IL-6 and CRP (ρ = −0.446, *p* = 0.017), as well as positive moderate correlation between IL-1 and SOD (ρ = 0.401, *p* = 0.034), were obtained preoperatively (Figure 4). Positive moderate correlation between MDA and CRP (ρ = 0.522, *p* = 0.004) and SOD and CRP (ρ = 0.452, *p* = 0.016) were shown on the 1st postoperative day (Figure 5). On the 7th postoperative day, SOD and CRP were in negative moderate correlation (ρ = −0.439, *p* = 0.019), while GPX and MDA (ρ = 0.403, *p* = 0.033) and GPX and IL-6 (ρ = 0.437, *p* = 0.020) were in positive moderate correlation (Figure 6).

### 3.6. Factors Influencing the Level of Oxidative Stress

We further evaluated factors influencing the level of oxidative stress (SOD and MDA) on the 1st and 7th postoperative days. The results of regression modeling are presented in Table 2. Factors correlated with higher SOD levels on the 1st postoperative day were higher pain intensity and receiving opioids (tramadol) on the same postoperative day, while, after multivariate linear regression, receiving opioids (tramadol) on the same postoperative day remained significantly correlated with higher levels of SOD (B = 40.510, 95%CI B = 23.73–57.29, *p* < 0.001). Possible factors correlating with higher SOD levels on the 7th postoperative day were tumor grade 0, T1, higher levels of CRP, and receiving opioids (tramadol) on the 1st postoperative day, while CRP on the 1st postoperative day remained significantly correlated with higher levels of SOD levels on the 7th postoperative day (B = 40.510, 95%CI B = 23.73–57.29, *p* < 0.001).

The only factor significantly associated with higher MDA levels on the 1st postoperative day was higher CRP (B = 18.927, 95%CI B = 6.01–31.85, *p* = 0.006), while older age and types of laryngectomy other than partial were independently associated with higher levels of MDA on the 7th postoperative day (B = 1.680, 95%CI B = 0.36–3.00, *p* = 0.015 and B = −25.919, 95%CI B = −48.23 to −3.60, *p* = 0.025, respectively) (Table 3).

## 4. Discussion

This is the first study where correlation between oxidative stress and inflammation biomarkers to postoperative pain modulation was examined and established in surgically treated patients with laryngeal cancer. Only a small number of studies have investigated the effectiveness of postoperative pain control in patients with laryngeal cancer. In the systematic review published by van den Beuken-van Everdingen et al., prevalence of pain in cancer patients was higher than 50%, and the highest prevalence was found in the head and neck cancer patients (70%) [26]. Study on incidence of postoperative pain in patients undergoing surgery for malignant head and neck disease shows that 48% of patients present a pain intensity greater than 4 on the VAS [27]. Most of the data from the literature indicate that management of acute postoperative pain is still sub-optimal, and that only 35% of post-laryngectomy patients received adequate and effective pain management [28,29]. Inadequate postoperative pain management was correlated with a prolonged hospital stay, immunosuppression, patient immobility, and increased morbidity [30].

In most otorhinolaryngology centers, opioid analgesics are a major component of the postoperative pain control plan. Most patients are managed with opioid intravenous patient-controlled analgesia (PCA) for the first five postoperative days and are switched to a combination of opioid and non-opioid analgesics (acetaminophen and/or ibuprofen) [31]. According to the VAS, patients in our study assessed their postoperative pain from mild to moderate (the VAS varied from 10–60). Mild pain was managed with one or a combination of two non-opioid analgesics (acetaminophen and NSAIDs) in 16 (57.1%) patients. Opioid analgesic (tramadol) alone or in a combination with NSAIDs was used in 12 (42.9%) patients with moderate pain. The quantity of analgesics and the duration of the therapy in days were more dependent on VAS scores. Only 11 patients used multimodal analgesia (39.3%). The length of usage of analgesics was individual, and it varied from 3 to 7 days. Increase of VAS scores in the intermediate postoperative course in our study could indicate that pain management was not adequate, since data in the literature shows that better pain control is obtained with opioids alone or in combination with other analgesics (paracetamol and/or non-steroidal anti-inflammatory drugs) in the first 2 days after surgery [31]. Recently, Dort et al. [32] presented an enhanced recovery after surgery (ERAS) guideline for head and neck cancer patients which used multimodal analgesia (MMA) to accomplish effective pain management. MMA is defined as the concurrent use of more than one modality of pain control to achieve effective analgesia. MMA was effective in shortening hospital stays, decreasing morbidity, and reducing opioid consumption and opioid-related side effects. MMA protocols provide early improvement of postoperative pain in patients undergoing major head and neck surgery [33,34,35]. In the tertiary otorhinolaryngology medical center where our study was conducted, opioids are mostly used with caution, in managing moderate and severe pain in patients postoperatively and in palliative pain management. Along with risk of dependence, the adverse effects associated with opioid medications include nausea and vomiting, constipation, impaired mobilization, sedation, and delirium, especially in older patients with comorbidities [31,32]. Still, opioid use guarantees better postoperative pain control, especially with pain of higher intensity. It is our experience from this study that MMA which includes use of opioids with NSAIDs for a limited period of time in the immediate postoperative course (3 to 5 days) could be the most optimal pain management regime.

During and after surgery, the body undergoes significant physiological stress, which activates the hypothalamic–pituitary secretion axis, increases secretion of pro-inflammatory factors, and results in hyperalgesia and aggravation of inflammatory response. Surgical procedures cause an increase in ROS production and a decrease in antioxidant defense mechanisms, leading to oxidative stress [36]. MDA is a commonly used biomarker for oxidative stress and lipid peroxidation in biological systems and is considered a pro-oxidant [37]. On the other hand, superoxide dismutase (SOD) and glutathione peroxidase 1 (GPX1) are antioxidant enzymes. They behave as scavengers of free radicals and provide the first line of defense against tissue damage caused by ROS [6]. The association of oxidative stress and inflammation parameters with postoperative pain after various surgical interventions has been the subject of research in the last few years. This association has not yet been examined in surgically treated patients with laryngeal cancer.

In this study, levels of MDA significantly increased on the 1st postoperative day and remained high on the 7th postoperative day, marking the development and maintenance of oxidative stress postoperatively. Similarly, levels of SOD significantly increased on the 1st postoperative day and remained high on the 7th postoperative day, indicating that anti-oxidative protection was also increased. Levels of GPX were elevated postoperatively, but without a statistical difference. In the literature, MDA serum levels were significantly increased postoperatively in patients after cardiothoracic, orthopedic, and ophthalmologic surgery [38,39,40,41]. Similar to our results, Kärkkäinen et al. [8] found that SOD plasma concentrations significantly increased in the first 24 h after abdominal operations involving laparotomy. This study also highlighted a significant positive correlation between SOD and GPX in the postoperative period and found significant negative correlation between SOD and the subjectively assessed level of postoperative pain. In a study conducted by Liu et al. in the pediatric population undergoing appendectomy, plasma SOD concentrations were observed to be elevated in the postoperative period compared to a control non-operated group [9]. Salehi et al. found elevated SOD and GPX levels in patients after colorectal adenocarcinoma surgery [42]. Our results indicate that, despite the existence of oxidative stress shown by elevated values of MDA postoperatively, the antioxidant protection system was activated through increased levels of SOD and GPX.

Our study detected some increase of serum IL-1 and IL-6 values in the postoperative period, but the difference was not significant. Additionally, there was not any significant correlation between VAS scores and IL-1 and IL-6 values postoperatively. The literature data on the correlation between serum values of pro-inflammatory cytokines and postoperative pain intensity varies. Some studies found positive correlation between serum pro-inflammatory cytokines (IL-6, IL-8 and TNF-α) and the intensity of acute postoperative pain [17,43]. Increased levels of IL-6 and IL-8 were detected in the first 24 h after the surgery [44], and their elevated values still persisted for a week in recovering patients [43]. Other authors did not find correlation between postoperative pain intensity and serum concentrations of pro-inflammatory cytokines (tumor necrosis factor α, TNF α, and interleukin-6) [45]. It was shown that in cases of better-controlled postoperative analgesia, values of pro-inflammatory cytokines (IL-6) and oxidative stress factor (MDA) were reduced and plasma SOD concentration was higher [46,47]. While we detected positive moderate correlation between MDA and IL-6 and negative moderate correlation between SOD and CRP in the postoperative period, the reported VAS scores and parameters of oxidative stress showed no significant correlation. These results could be explained by the relatively small number of patients involved in the cited studies, including ours. Additionally, the types of surgery performed are heterogeneous and involve surgical treatment on different organ systems, which all have their specificity and could influence the outcomes. These results could be subject to change after evaluating a higher number of samples in oncological head and neck surgery.

We established that higher CRP values were predictors of higher MDA levels on the 1st postoperative day, while older age and more extensive surgery were predictors of higher levels of MDA on the 7th postoperative day. More extensive surgical tissue damage causes significant inflammation and ischemia/reperfusion injury, and ROS are important mediators of this damage [48]. The extent and the technique of the surgical intervention have an impact on the intraoperative oxidative stress response and postoperative recovery. Compared to less invasive interventions, higher levels of oxidative stress have been found in more complex surgeries [47]. Additionally, oxidative stress can have a particularly negative impact in all forms of major surgery in an ageing population, since older age was associated with decreased levels of antioxidant mechanisms and increased oxidative damage in the previous studies [49].

In our study, use of tramadol for pain management was a predictive factor of higher SOD levels on the 1st postoperative day, and lower CRP values were a predictive factor for higher SOD values on the 7th postoperative day. Increased levels of SOD in patients who used tramadol for postoperative pain management could indicate that tramadol stimulates activation of the antioxidant protection system. Tramadol is an opioid analgesic used to treat moderate to severe pain caused by cancer and rheumatic and musculoskeletal diseases. This specific opioid analgesic has a unique mechanism of action among all opioids: binding to the μ-opioid receptors and inhibition of the neuronal uptake of the neurotransmitters norepinephrine and serotonin [50]. It was established that tramadol acts as an anti-inflammatory agent. Tramadol can suppress the cytokine storm through decreasing interleukins such as IL6, tumor necrosis factor-alpha (TNF-α), and C-reactive protein (CRP) [51,52]. Tramadol can also increase levels of the antioxidant enzymes superoxide dismutase and glutathione peroxidase and decrease the level of MDA in testicular ischemia–reperfusion injury [53]. These results could indicate that tramadol might be used as a potent antioxidant agent, as well as an integral part of optimal postoperative analgesia in patients with laryngeal cancer.

There are a few strengths of this study. First, for the first time, parameters of oxidative stress were correlated with VAS score in surgically treated patients with laryngeal carcinoma. Secondly, we established a significant correlation between tramadol use and increased levels of SOD and, therefore, a potential new correlation between tramadol and parameters of oxidative stress after surgical interventions. The limitations of our study include a small sample; larger number of patients are required to support our conclusion.

## 5. Conclusions

There is correlation between oxidative stress and inflammation biomarkers to postoperative pain modulation in surgically treated patients with laryngeal cancer. Age, more extensive surgery, CRP values, and use of tramadol were predictors for oxidative stress parameters. However, as this is a pilot study, further studies with a larger number of patients are required to support our conclusion.

## Figures and Tables

**Figure 1 cells-12-01391-f001:**
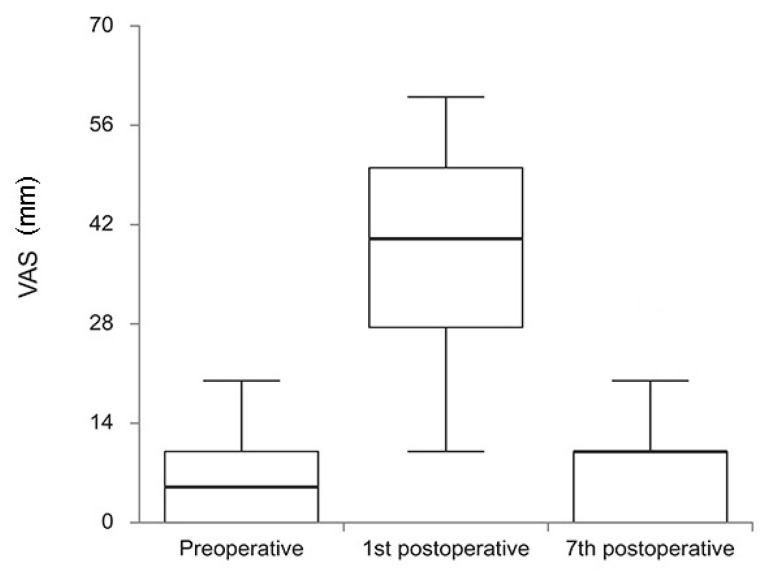
Visual analog scale (VAS) scores of the patients through the evaluated period of time (preoperative, 1st postoperative, and 7th postoperative day). There was a statistically significant difference between pain intensity preoperative and on the 1st postoperative day, as well as from the 1st to the 7th postoperative day (*p* < 0.001).

**Figure 2 cells-12-01391-f002:**
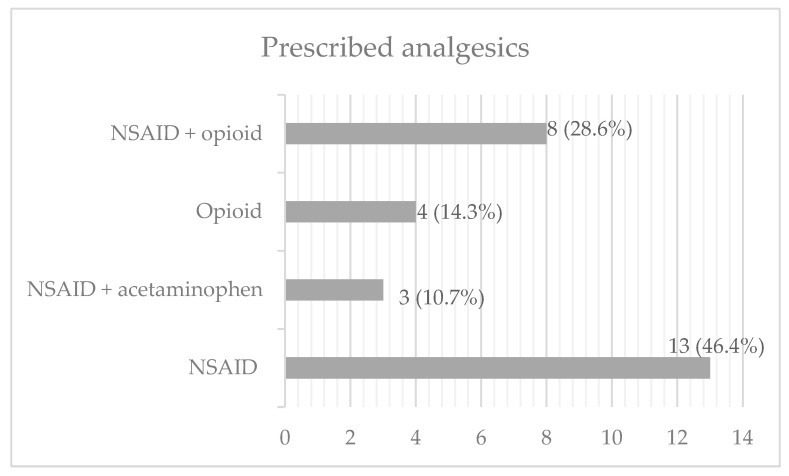
Analgesics prescribed in pain therapy. Ketorolac or metamizole were used as NSAIDs. Tramadol was used as the only opioid analgesic. Multimodal analgesia involved the use of NSAIDs and an opioid or NSAIDs and acetaminophen.

**Figure 3 cells-12-01391-f003:**
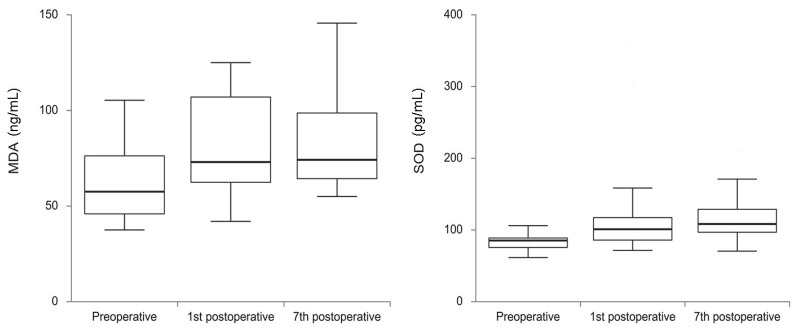
Values of oxidative stress parameters through the evaluated period of time. There was a significant change in values of SOD as a parameter of anti-oxidative stress and MDA as a parameter of oxidative stress from preoperative to the 7th postoperative day (*p* < 0.001). (*MDA*—*malondialdehyde*; *SOD*—*superoxide dismutase*).

**Figure 4 cells-12-01391-f004:**
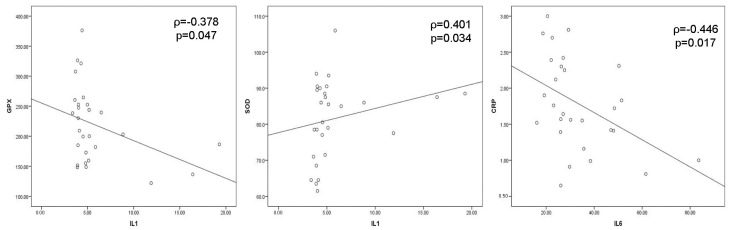
Significant correlations between oxidative stress and inflammation parameters preoperatively. Negative moderate correlation between IL-1 and GPX and between IL-6 and CRP and positive moderate correlation between IL-1 and SOD were obtained for the level of significance of <0.05 according to Spearman’s rank correlation coefficient (*GPX*—*glutathione peroxidase*, *SOD*—*superoxide dismutase*, *IL-1*—*interleukin 1*, *IL-6*—*interleukin-6*, *CRP*—*C-reactive protein*).

**Figure 5 cells-12-01391-f005:**
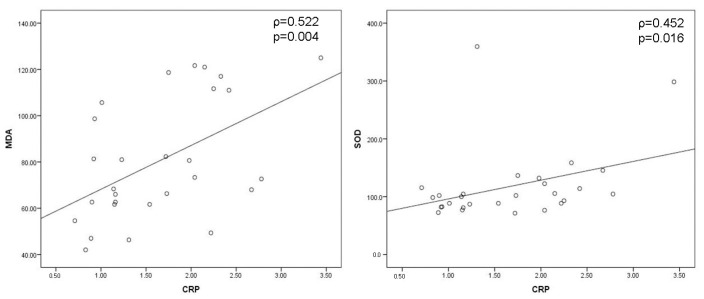
Significant correlations between oxidative stress and inflammation parameters on the 1st postoperative day. Positive moderate correlation between MDA and CRP and SOD and CRP were shown for the level of significance of <0.05 according to Spearman’s rank correlation coefficient (*MDA*—*malondialdehyde*, *SOD*—*superoxide dismutase*, *CRP*—*C-reactive protein*).

**Figure 6 cells-12-01391-f006:**
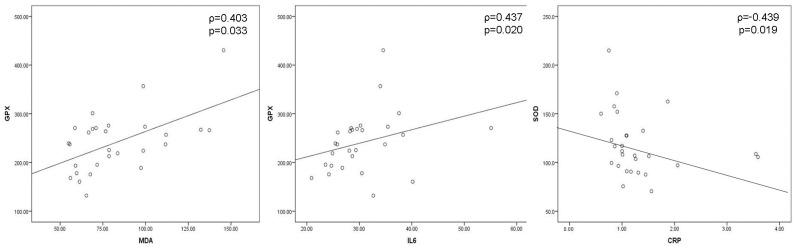
Significant correlations between oxidative stress and inflammation parameters on the 7th postoperative day. Negative moderate correlation between SOD and CRP and positive moderate correlation between GPX and MDA and GPX and IL-6 were shown for the level of significance of <0.05 according to Spearman’s rank correlation coefficient (*MDA*—*malondialdehyde*, *GPX*—*glutathione peroxidase*, *SOD*—*superoxide dismutase*, CRP—*C-reactive protein*).

**Table 1 cells-12-01391-t001:** Patients’ demographic and clinical characteristics.

Characteristic	
**Age (years), $x^{¯}$ ± sd**	67.04 ± 6.60
**Gender, *n* (%)**	
Males	23 (82.1)
Females	5 (17.9)
**Smoking status, *n* (%)**	
Smoker	24 (85.7)
Non-smoker	4 (14.3)
**Histological grade, *n* (%)**	
In situ	3 (10.7)
G1	4 (14.3)
G2	20 (71.4)
G3	3 (3.6)
**T stage *n* (%)**	
T1	9 (32.1)
T2	6 (21.4)
T3	8 (28.6)
T4	5 (17.9)
**N stage *n* (%)**	
N0	23 (82.2)
N1	1 (3.6)
N2	2 (7.1)
N3	2 (7.1)
**Type of surgical intervention, *n* (%)**	
Cordectomy	10 (35.7)
Partial laryngectomy	5 (17.9)
Total laryngectomy	8 (28.6)
Laryngectomy with neck dissection	5 (17.9)

**Table 2 cells-12-01391-t002:** Factors influencing the level of oxidative stress measured by SOD on the 1st and 7th postoperative day.

Factor	B	95%CI B	*p*
**1st postoperative day**			
VAS pain intensity	1.104	−0.14–2.35	0.081
Opioid (tramadol) vs. combination of analgesics (opioids+NSAIL) on the 1st postoperative day	40.510	23.73–57.29	<0.001 *
**7th postoperative day**			
Tu in situ vs. grades 1, 2, 3	28.511	−6.29–63.31	0.104
T1 vs. T2, T3, T4	−5.334	−31.47–20.80	0.677
CRP	22.849	5.29–40.40	0.013 *
Opioid (tramadol) vs. combination of analgesics (opioids+NSAID) on the 1st postoperative day	7.049	−2.54–16.64	0.142

* for the level of significance of <0.05.

**Table 3 cells-12-01391-t003:** Factors influencing the level of oxidative stress measured by MDA on the 1st and 7th postoperative day.

Factor	B	95%CI B	*p*
**1st postoperative day**			
CRP	18.927	6.01–31.85	0.006
**7th postoperative day**			
Age	1.680	0.36–3.00	0.015
Partial laryngectomy vs. other	−25.919	−48.23 to −3.60	0.025

## Data Availability

Data will be available on request from the corresponding author (anajotic@yahoo.com).

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
