# Peer review of "Oxidative Stress and Inflammation Biomarkers in Postoperative Pain Modulation in Surgically Treated Patients with Laryngeal Cancer—Pilot Study"

_cells, 2023, doi:10.3390/cells12101391_

Round 1
Reviewer 1 Report
The authors did an interesting work, but I am concerned on the novelty that the paper brings. The results may be interesting, but I am not sure if it is strong enough to be published on the journal, or at least as the way they are represented.
On methods section the postoperative analgesia topic must be rewritten with a clearer way. The text is confusing on describing the drug treatments.
Table 1 must be reorganized to a better representation because it is too long with only 2 columns. There must be a better way to represent this data.
Can’t tables 2 and 3 be transformed into graphic?
Figure 1 and its description is confusing, it is hard to understand the graphic and the description you did on the text. Maybe because the way you chose to represent the treatments. So, this must be better organized and described to avoid data misunderstanding. The treatments here must agree with the methods section that must be changed.
The English quality is good, there are only a few typos mistakes that must be carefully corrected
Author Response
We would like to thank the reviewer on their time to review the manuscript and on their very useful and constructive comments. It allowed us to better clarify and represent some parts of the Material and methods and Results section, which made the manuscript better and more concise.
- The authors did an interesting work, but I am concerned on the novelty that the paper brings. The results may be interesting, but I am not sure if it is strong enough to be published on the journal, or at least as the way they are represented.
Response: We rewritten some part of the Results and Discussion section, which gave clarity to presentation of our findings in the paper. Also, results of our study were further explained and their novelty of further enhanced in the Discussion section. We hope that the changes we made will justify publishing first study that deals with oxidative stress and postoperative management in patients with laryngeal cancer.
- On methods section the postoperative analgesia topic must be rewritten with a clearer way. The text is confusing on describing the drug treatments.
Response: Section on postoperative analgesia was rewritten and some parts of the Material and method section reorganized. We defined what analgesics were used in patients with mild and moderate pain (according to the recommendations cited in the article) and how the re-assessment was made in pain control wasn’t achieved.
- Table 1 must be reorganized to a better representation because it is too long with only 2 columns. There must be a better way to represent this data.
Response: Other solutions for the table 1 were tried, but more than two columns were too confusing for presenting these data. We think the look of this table is the most optimal solution for presenting demographic and clinical characteristics of the patients included in the study.
- Can’t tables 2 and 3 be transformed into graphic?
Response: Data from tables 2 and 3 were transformed into graphs (figure 1, 3, 4, 5 and 6) for better data clarity. Changes in the text in the Results section were also made.
- Figure 1 and its description is confusing, it is hard to understand the graphic and the description you did on the text. Maybe because the way you chose to represent the treatments. So, this must be better organized and described to avoid data misunderstanding. The treatments here must agree with the methods section that must be changed.
Response: Section on postoperative analgesia was rewritten in the Material and methods section, and further corrected in the Results section. Some small changes were made to the figure 1 (figure 2 in the new version of the manuscript).
- Comments on the Quality of English Language-The English quality is good, there are only a few typos mistakes that must be carefully corrected.
Response: The manuscript was type-checked, and mistakes corrected.
Note to the reviewer: The journal recommends a minimum word count of 4000, so we extended the manuscript to meet the journal requirements. All changes in the manuscript were noted in track changes and colored red in
Reviewer 2 Report
Particularly, I listed the following comments in detail here.
In the abstract, the author needs to mention the ingredients of methods and materials. Also, the finding of the assay could be added step by step based on material and method.
In the introduction, some sentences lack references, for example, “Surgical treatment of laryngeal carcinoma includes different types of more or less extensive 42
laryngectomies with reconstructions of the aerodigestive tract with local or remote flaps, 43
and associated unilateral or bilateral neck dissection.” “The process of oxidative stress causes the peroxidation of lipids and proteins, the formation of lipid peroxides, DNA fragmentation and the development of cell death. Malondialdehyde (MDA) is a breakdown product of lipid peroxides, belongs to the group of aldehydes and is used for the quantification of lipid peroxidation in the obtained tissue samples.”, and so on.
In the discussion, discuss your results before relating them to the results of other published work. Also, the author must step by step to come to the results and comparison with others. What is your conclusion? Hence, add a significant statement that must be structured as “What was offered by authors? Do the authors have more thoughts on this field?
English writing can be improved.
Author Response
We would like to thank the reviewer on their time to review the manuscript and on their helpful comments in order to further improve the manuscript.
- In the abstract, the author needs to mention the ingredients of methods and materials. Also, the finding of the assay could be added step by step based on material and method.
Response: We added the part referring the assay used for determining the serum concentration of oxidative stress and inflammation parameters. Due to a limited word count (only 200 words allowed for the abstract) we were not been able to explain in detail methods used in the study.
- In the introduction, some sentences lack references, for example, “Surgical treatment of laryngeal carcinoma includes different types of more or less extensive 42
laryngectomies with reconstructions of the aerodigestive tract with local or remote flaps, 43
and associated unilateral or bilateral neck dissection.” “The process of oxidative stress causes the peroxidation of lipids and proteins, the formation of lipid peroxides, DNA fragmentation and the development of cell death. Malondialdehyde (MDA) is a breakdown product of lipid peroxides, belongs to the group of aldehydes and is used for the quantification of lipid peroxidation in the obtained tissue samples.”, and so on.
Response: The appropriate reference were added in the manuscript, and in the literature section.
- In the discussion, discuss your results before relating them to the results of other published work. Also, the author must step by step to come to the results and comparison with others. What is your conclusion? Hence, add a significant statement that must be structured as “What was offered by authors? Do the authors have more thoughts on this field?
Response: Parts of the Discussion section were rewritten, some explanations added on the how our results compare to other available and what the differences mean from our standpoint. Appropriate references were added.
- Comments on the Quality of English Languag-English writing can be improved
Response: the manuscript was revised by a English native speaker, and hopefully any mistakes were addressed.
Note to the reviewer: The journal recommends a minimum word count of 4000, so we extended the manuscript to meet the journal requirements.
Round 2
Reviewer 1 Report
The authors reply to al my suggestions and made a big increase on the paper quality and writing.
The only thing I will ask is to provide a description of the figure legends. They are a little bit poor with only the tittle line. A small description needs to be added on each figure legend.
After this the paper can be accepted.
Author Response
The authors reply to al my suggestions and made a big increase on the paper quality and writing.
Response: Thank you again for your constructive comments. They made a huge difference, and the paper is now better written and more focused on the subject.
The only thing I will ask is to provide a description of the figure legends. They are a little bit poor with only the tittle line. A small description needs to be added on each figure legend.
Response: We added the figure legend on the every figure, which is noted in the new version of the manuscript.
After this the paper can be accepted.